# Meta-Surrogate Benchmarking for Hyperparameter Optimization

**Aaron Klein**[1]     **Zhenwen Dai**[2]     **Frank Hutter**[1]     **Neil Lawrence**[3]     **Javier González**[2]

[1]University of Freiburg     [2]Amazon Cambridge     [3]University of Cambridge

{kleinaa,fh}@cs.uni-freiburg.de
{zhenwend, gojav}@amazon.com
ndl21@cam.ac.uk

## Abstract

Despite the recent progress in hyperparameter optimization (HPO), available benchmarks that resemble real-world scenarios consist of a few and very large problem instances that are expensive to solve. This blocks researchers and practitioners not only from systematically running large-scale comparisons that are needed to draw statistically significant results but also from reproducing experiments that were conducted before. This work proposes a method to alleviate these issues by means of a meta-surrogate model for HPO tasks trained on off-line generated data. The model combines a probabilistic encoder with a multi-task model such that it can generate inexpensive and realistic tasks of the class of problems of interest. We demonstrate that benchmarking HPO methods on samples of the generative model allows us to draw more coherent and statistically significant conclusions that can be reached orders of magnitude faster than using the original tasks. We provide evidence of our findings for various HPO methods on a wide class of problems.

## 1   Introduction

Automated Machine Learning (AutoML) [19] is an emerging field that studies the progressive automation of machine learning. A core part of an AutoML system is the hyperparameter optimization (HPO) of a machine learning algorithm. It has already shown promising results by outperforming human experts in finding better hyperparameters [34], an thereby, for example, substantially improved AlphaGo [7].

Despite recent progress (see e. g.  the review by Feurer and Hutter [11]), during the phases of developing and evaluating new HPO methods one frequently faces the following problems:

- Evaluating the objective function is often expensive in terms of wall-clock time; *e.g.*, the evaluation of a single hyperparameter configuration may take several hours or days. This renders extensive HPO or repeated runs of HPO methods computationally infeasible.

- Even though repositories of datasets, such as OpenML [41] provide thousands of datasets, a large fraction cannot meaningfully be used for HPO since they are too small or too easy (in the sense that even simple methods achieve top performance). Hence, useful available datasets are scarce, making it hard to produce a comprehensive evaluation of how well a HPO method will generalize across tasks.

Due to these two problems researchers can only carry out a limit number of comparisons within a reasonable computational budget. This delays the progress of the field as statistically significant conclusions about the performance of different HPO methods may not be possible to draw. See Figure 1 for an illustrative experiment of the HPO of XGBoost [4]. It is well known that Bayesian

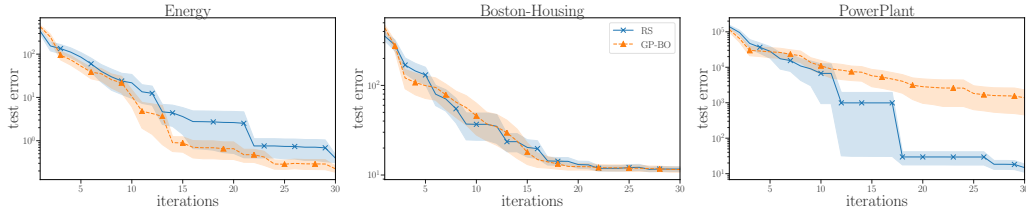

Figure 1: Common pitfalls in the evaluation of HPO methods: we compare two different HPO methods for optimizing the hyperparameters of XGBoost on three UCI regression datasets (see Appendix B for more datasets). The small number of tasks makes it hard to draw any conclusions, since the ranking of the methods varies between tasks. Furthermore, a full run might take several hours which makes it prohibitively expensive to average across a large number of runs.

optimization with Gaussian processes (BO-GP) [33] outperforms naive random search (RS) in terms of number of function evaluations on most HPO problems. While we show clear evidence for this in Appendix B on a larger set of datasets, this conclusion cannot be reached when optimizing on the three "unlucky" picked datasets in Figure 1. Surprisingly, the community has not paid much attention to this issue of proper benchmarking, which is a key step required to generate new scientific knowledge.

In this work we present a generative meta-model that, conditioned on off-line generated data, allows to sample an unlimited number of new tasks that share properties with the original ones. There are several advantages to this approach. First, the new problem instances are inexpensive to evaluate as they are generated with a parameteric form, which drastically reduces the resources needed to compare HPO methods, bounded only by the optimizer's computational overhead (see Figure 2 for an example). Second, there is no limit in the number of tasks that can be generated, which helps to draw statistically more reliable conclusions. Third, the shape and properties of the tasks are not predefined but learned using a few real tasks of an HPO problem. While the *global* properties of the initial tasks are preserved in the samples, the generative model allows the exploration of instances with diverse *local* properties making comparisons more robust and reliable (see Appendix D for some example tasks).

In light of the recent call for more reproducibility, we are convinced that our meta-surrogate bench-

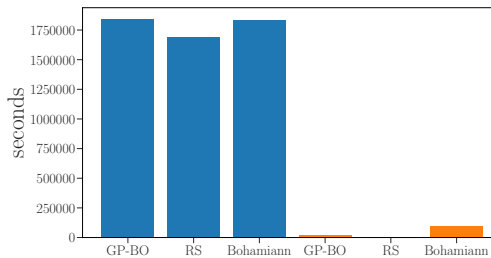

Figure 2: The three blue bars on the left show the total wall-clock time of executing 20 independent runs of GP-BO, RS and Bohamiann (see Section 5) with 100 function evaluations for the HPO of a feed forward neural network on MNIST. The orange bars show the same for optimizing a task sampled from our proposed meta-model, where benchmarking is orders of magnitude cheaper in terms of wall-clock time than the original benchmarks, thereby the computational time is almost exclusively spend for the optimizer (hence the larger bars for GP-BO and Bohamiann compared to RS).

marks enable more reproducible research in AutoML: First of all, these cheap-to-evaluate surrogate benchmarks allows researches to reproduce experiments or perform many repeats of their own experiments without relying on tremendous computational resources. Second, based on our-proposed method, we provide a more thorough benchmarking protocol that reduces the risk of extensively tuning an optimization method on single tasks. Third, surrogate benchmarks in general are less dependent on hardware and technical details, such as complicated training routines or preprocessing strategies.

## 2   Related Work

The use of meta-models that learn across tasks has been investigated by others before. To warm-start HPO on new tasks from previously optimized tasks, Swersky et al. [38] extended Bayesian optimization to the multi-task setting by using a Gaussian process that also takes the correlation between tasks into account. Instead of a Gaussian process, Springenberg et al. [36] used a Bayesian neural network inside multi-task Bayesian optimization which learns an embedding of tasks during optimization.

Similarly, Perrone et al. [32] used Bayesian linear regression, where the basis functions are learned by a neural network, to warm-start the optimization from previous tasks. Feurer et al. [13] used a set of dataset statistics as meta-features to measure the similarity between tasks, such that hyperparameter configurations that were superior on previously optimized similar tasks can be evaluated during the initial design. This technique is also applied inside the auto-sklearn framework [12]. In a similar vein, Fusi et al. [14] proposed to use a probabilistic matrix factorization approach to exploit knowledge gathered on previously seen tasks. van Rijn and Hutter [40] evaluated random hyperparameter configurations on a large range of tasks to learn priors for the hyperparameters of support vector machines, random forests and Adaboosts. The idea of using a latent variable to represent correlation among multiple outputs of a Gaussian process has been exploited by Dai et al. [8].

Besides the aforementioned work on BO, also other methods have been proposed for efficient HPO. Li et al. [29] proposed Hyperband, which based on the bandit strategy successive halving [21], dynamically allocates resources across a set of random hyperparameter configurations. Similarly, Jaderberg et al. [20] presented an evolutionary algorithm, dubbed PBT, which adapts a population of hyperparameter configurations during training by either random perturbations or exploiting values of well-performing configurations in the population.

In the context of benchmarking HPO methods, HPOlib [9] is a benchmarking library that provides a fixed and rather small set of common HPO problems. In earlier work, Eggensperger et al. [10] proposed surrogates to speed up the empirical benchmarking of HPO methods. Similar to our work, these surrogates are trained on data generated in an off-line step. Afterwards, evaluating the objective function only requires querying the surrogate model instead of actually running the benchmark. However, these surrogates only mimic one particular task and do not allow for generating new tasks as presented in this work. Recently, tabular benchmarks were introduced for neural architecture search [43] and hyperparameter optimization [24], which first perform an exhaustive search of a discrete benchmark problem to store all results in a database and then replace expensive function evaluations by efficient table lookups. While this does not introduce any bias due to a model (see Section 6 for a more detailed discussion), tabular benchmarks are only applicable for problems with few, discrete hyperparameters. Related to our work, but for benchmarking general blackbox optimization methods, is the COCO platform [17]. However, compared to our approach, it is based on handcrafted synthetic functions that do not resemble real world HPO problems.

## 3 Benchmarking HPO methods with generative models

We now describe the generative meta-model to create HPO tasks. First we give a formal definition of benchmarking HPO methods across tasks sampled from a unknown distribution and then describe how we can approximate this distribution by our new proposed meta-model.

### 3.1 Problem Definition

We denote $t_1, \ldots, t_M$ to be a set of related objectives/tasks with the same input domain $\mathcal{X}$, for example $\mathcal{X} \subset \mathbb{R}^d$. We assume that each $t_i$ for $i = 1, \ldots M$, is an instantiation of an unknown distribution of tasks $t_i \sim p(t)$. Every task $t$ has an associated objective function $f_t : \mathcal{X} \to \mathbb{R}$ where $\boldsymbol{x} \in \mathcal{X}$ represents a hyperparameter configuration and we assume that we can observe $f_t$ only through noise: $y_t \sim \mathcal{N}(f_t(\boldsymbol{x}), \sigma_t^2)$.

Let us denote by $r(\alpha, t)$ the performance of an optimization method $\alpha$ on a task $t$; for instance, a common example for $r$ is the regret of the best observed solution (called incumbent). To compare two different methods $\alpha_A$ and $\alpha_B$, the standard practice is to compare $r(\alpha_A, t_i)$ with $r(\alpha_B, t_i)$ on a set of hand-picked tasks $t_i \in \{t_1, \ldots t_M\}$. However, to draw statistically more significant conclusion, we would ideally like to integrate over all tasks:

$$S_{p(t)}(\alpha) = \int r(\alpha, t) p(t) dt. \tag{1}$$

Unfortunately, the above integral is intractable as $p(t)$ is unknown. The main contribution of this paper is to approximate $p(t)$ with a generative meta-model $\hat{p}(t \mid \mathcal{D})$ based on some off-line generated data $\mathcal{D} = \left\{ \{(\boldsymbol{x}_{tn}, y_{tn})\}_{n=1}^N \right\}_{t=1}^T$. This enables us to sample an arbitrary amount of tasks $t_i \sim \hat{p}(t \mid \mathcal{D})$ in order to perform a Monte-Carlo approximation of Equation 1.

## 3.2 Meta-Model for Task Generation

In order to reason across tasks, we define a probabilistic encoder $p(\mathbf{h}_t \mid \mathcal{D})$ that learns a latent representation $\mathbf{h}_t \in \mathbb{R}^Q$ of a task $t$.

More precisely, we use Bayesian GP-LVM [39] which assumes that the target values that belong to the task $t$, stacked into a vector $\boldsymbol{y}_t = (y_{t1}, \ldots, y_{tN})$ follow the generative process:

$$\boldsymbol{y}_t = g(\mathbf{h}_t) + \epsilon, \quad g \sim \mathcal{GP}(0, k), \quad \epsilon \sim \mathcal{N}(0, \sigma^2), \tag{2}$$

where $k$ is the covariance function of the GP. By assuming that the latent variable $\mathbf{h}_t$ has an uninformative prior $\mathbf{h}_t \sim \mathcal{N}(0, \mathbf{I})$, the latent embedding of each task is inferred as the posterior distribution $p(\mathbf{h}_t \mid \mathcal{D})$. The exact formulation of the posterior distribution is intractable, but following the variational inference presented in Titsias and Lawrence [39], we can estimate a variational posterior distribution $q(\mathbf{h}_t) = \mathcal{N}(m_t, \Sigma_t) \approx p(\mathbf{h}_t \mid \mathcal{D})$ for each task $t$.

Similar to Multi-Task Bayesian Optimization [38, 36], we define a probabilistic model for the objective function $p(y_t \mid \boldsymbol{x}, \mathbf{h}_t)$ across tasks which gets as an additional input a task embedding based on our independently trained probabilistic encoder. Following Springenberg et al. [36], we use a Bayesian neural network with $M$ weight vectors $\{\boldsymbol{\theta}_1, \ldots, \boldsymbol{\theta}_M\}$ to model

$$p(y_t \mid \boldsymbol{x}, \mathbf{h}_t, \mathcal{D}) = \int p(y_t \mid \boldsymbol{x}, \mathbf{h}_t, \boldsymbol{\theta}) p(\theta \mid \mathcal{D}) d\boldsymbol{\theta} \quad \approx \frac{1}{M} \sum_{i=1}^{M} p(y_t \mid \boldsymbol{x}, \mathbf{h}_t, \boldsymbol{\theta}_i). \tag{3}$$

where $\boldsymbol{\theta}_i \sim p(\boldsymbol{\theta} \mid \mathcal{D})$ is sampled from the posterior of the neural network weights.

By approximating $p(y_t \mid \boldsymbol{x}, \mathbf{h}_t) = \mathcal{N}\big(\mu(\boldsymbol{x}, \mathbf{h}_t), \sigma^2(\boldsymbol{x}, \mathbf{h}_t)\big)$ to be Gaussian [36], we can compute the predictive mean and variance by:

$$\mu(\boldsymbol{x}, \mathbf{h}_t) = \frac{1}{M} \sum_{i=1}^{M} \hat{\mu}(\boldsymbol{x}, \mathbf{h}_t \mid \boldsymbol{\theta}_i) \quad ; \quad \sigma^2(\boldsymbol{x}, \mathbf{h}_t) = \frac{1}{M} \sum_{i=1}^{M} \big(\hat{\mu}(\boldsymbol{x}, \mathbf{h}_t \mid \boldsymbol{\theta}_i) - \mu(\boldsymbol{x}, \mathbf{h}_t)\big)^2 + \hat{\sigma}_{\boldsymbol{\theta}_i}^2,$$

where $\hat{\mu}(\boldsymbol{x}, \mathbf{h}_t \mid \boldsymbol{\theta}_i)$ and $\hat{\sigma}_{\boldsymbol{\theta}_i}^2$ are the output of a single neural network with parameters $\boldsymbol{\theta}_i$[1]. To get a set of weights $\{\boldsymbol{\theta}_1, \ldots, \boldsymbol{\theta}_M\}$, we use stochastic gradient Hamiltonian Monte-Carlo [5] to sample $\boldsymbol{\theta}_i \sim p(\boldsymbol{\theta}, \mathcal{D})$ from:

$$p(\boldsymbol{\theta}, \mathcal{D}) = \frac{1}{N} \sum_{n=1}^{N} \frac{1}{H} \sum_{j=1}^{H} \log p(y_n \mid \boldsymbol{x}_n, \mathbf{h}_{nj})$$

with $N = |\mathcal{D}|$ the number of datapoints in our training set and $H$ the number of samples we draw from the latent space $\mathbf{h}_{tj} \sim q(\mathbf{h}_t)$.

## 3.3 Sampling New Tasks

In order to generate a new task $t_\star \sim \hat{p}(t \mid \mathcal{D})$, we need the associated objective function $f_{t_\star}$ in a parameteric form such that we can evaluate it later on any $\boldsymbol{x} \in \mathcal{X}$.

Given the meta-model above, we perform the following steps: (i) we sample a new latent task vector $\mathbf{h}_{t_\star} \sim q(\mathbf{h}_t)$; (ii) given $\mathbf{h}_{t_\star}$ we pick a random $\boldsymbol{\theta}_i$ from the set of weights $\{\boldsymbol{\theta}_1, \ldots \boldsymbol{\theta}_M\}$ of our Bayesian neural network and set the new task to be $f_{t_\star}(\boldsymbol{x}) = \hat{\mu}(\boldsymbol{x}, \mathbf{h}_{t_\star} \mid \boldsymbol{\theta}_i)$.

Note that using $f_{t_\star}(\boldsymbol{x})$ makes our new task unrealisticly smooth. Instead, we can emulate the typical noise appearing in HPO benchmarks by returning $y_{t_\star}(\boldsymbol{x}) \sim \mathcal{N}\big(\hat{\mu}(\boldsymbol{x}, \mathbf{h}_{t_\star} \mid \boldsymbol{\theta}_i), \hat{\sigma}_{\boldsymbol{\theta}_i}^2\big)$, which can be done at an insignificant cost.

## 4 Profet

We now present our PRObabilistic data-eFficient Experimentation Tool, called PROFET, a benchmarking suite for HPO methods (an open-source implementation is available here:

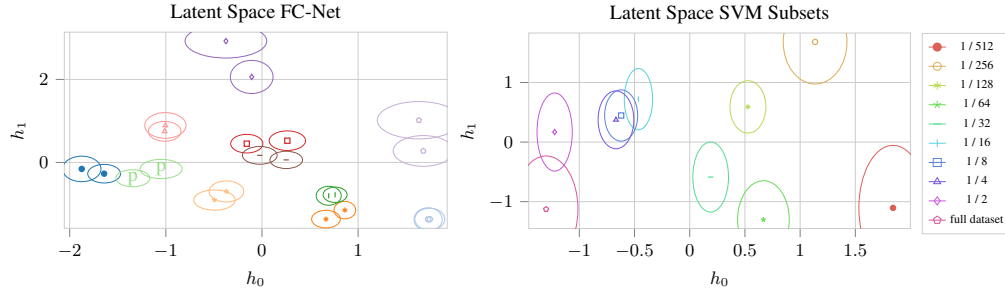

Figure 3: Latent space representations of our probabilistic encoder. *Left*: Representation of task pairs (same color) generated by partitioning eleven datasets from the fully connected network benchmark detailed in Section 4.1. Mean of tasks are visualized with different markers and ellipses represent 4 standard deviations. *Right*: Latent space learned for a model where the input tasks are generated by training a SVM on subsets of MNIST (see Klein et al. [25] for more details).

https://github.com/amzn/emukit). We provide pseudo code in Appendix G. The following section describes first how we collected the data to train our meta-model based on three typical HPO problems classes. We then explain how we generated $T = 1000$ different tasks for each problem class from our meta-model. As described above, we provide a noisy and noiseless version of each task. Last, we discuss two ways that are commonly used in the literature to assess and aggregate the performance of HPO methods across tasks.

## 4.1 Data Collection

We consider three different HPO problems, two for classification and one for regression, with varying dimensions $D$. For classification, we considered a support vector machine (SVM) with $D = 2$ hyperparameters and a feed forward neural network (FC-Net) with $D = 6$ hyperparameters on 16 OpenML [41] tasks each. We used gradient boosting (XGBoost)[2] with $D = 8$ hyperparameters for regression on 11 different UCI datasets [30]. For further details about the datasets and the configuration spaces see Appendix A. To make sure that our meta-model learns a descriptive representation we need a solid coverage over the whole input space. For that we drew $100D$ pseudo randomly generated configurations from a Sobol grid [35].

Details of our meta-model are described in Appendix F. We show some qualitative examples of our probabilistic encoder in Section 5.1. We can also apply the same machinery to model the cost in terms of computation time for evaluating a hyperparameter configuration to use time rather than function evaluations as budget. This enables future work to benchmark HPO methods that explicitly take the cost into account (e. g. EIperSec [34]).

## 4.2 Performance Assessment

To assess the performance of a HPO method aggregated over tasks, we consider two different ways commonly used in the literature. First, we measure the **runtime** $r(\alpha, t, y_{target})$ that a HPO method $\alpha$ needs to find a configuration that achieves a performance that is equal or lower than a certain target value $y_{target}$ on task $t$ [16]. Here we define runtime either in terms of function evaluations or estimated wall-clock time predicted by our meta-model. Using a fixed target approach allows us to make quantitative statements, such as: *method A is, on average, twice as fast than method B*. See Hansen et al. [16] for a more detailed discussion. We average across target values with a different complexity by evaluating the Sobol grid from above on each generated task. We use the corresponding function values as targets, which, with the same argument as described in Section 4.1, provides a good coverage of the error surface. To aggregate the runtime we use the empirical cumulative distribution function (ECDF) [31], which, intuitively, shows for each budget on the x-axis the fraction of solved tasks and target pairs on the y-axis (see Figure 5 left for an example).

Another common way to compare different HPO methods is to compute the **average ranking score** in every iteration and for every task [1]. We follow the procedure described by Feurer et al. [13] and compute the average ranking score as follows: assuming we run $K$ different HPO methods $M$ times

for each task, we draw a bootstrap sample of 1000 runs out of the $K^M$ possible combinations. For each of these samples, we compute the average fractional ranking (ties are broken by the average of the ordinal ranks) after each iteration. At the end, all the assigned ranks are further averaged over all tasks. Note that, averaged ranks are a relative performance measurement and can worsen for one method if another method improves (see Figure 5 right for an example).

## 5   Experiments

In this section we present: (i) some qualitative insights of our meta-model by showing how it is able to coherently represent a sets of tasks in its latent space, (ii) an illustration of why PROFET helps to obtain statistically meaningful results and (iii) a comparison of various methods from the literature on our new benchmark suite. In particular, we show results for the following state-of-the-art BO methods as well as two popular evolutionary algorithms:

- BO with Gaussian processes (BO-GP) [22]. We used expected improvement as acquisition function and marginalize over the Gaussian process' hyperparameters as described by Snoek et al. [34].
- SMAC [18]: which is a variant of BO that uses random forests to model the objective function and stochastic local search to optimize expected improvement.
  We use the implementation from https://github.com/automl/SMAC3.
- The BO method TPE by Bergstra et al. [3] which models the density of good and bad configurations in the input space with a kernel density estimators. We used the implementation provided from the Hyperopt package [28]
- BO with Bayesian neural networks (BOHAMIANN) as described by Springenberg et al. [36]. To avoid introducing any bias, we used a different architecture with less parameters (3 layers, 50 units in each) than we used for our meta-model (see Section 3).
- Differential Evolution (DE) [37] (we used our own implementation) with rand1 strategy for the mutation operators and a population size of 10.
- Covariance Matrix Adaption Evolution Strategy (CMA-ES) by Hansen [15] where we used the implementation from https://github.com/CMA-ES/pycma
- Random Search (RS) [2] which samples configurations uniformly at random.

For BO-GP, BOHAMIANN and RS we used the implementation provided by the RoBO package [26]. We provide more details for every method in Appendix E.

### 5.1   Tasks Representation in the Latent Space

We demonstrate the interpretability of the learned latent representations of tasks in two examples. For the first experiment we used the fully connected network benchmark described in Section 4.1. To visualize that our meta-model learns a meaningful latent space, we doubled 11 out of the 18 original tasks to train the model by splitting each one of them randomly in two of the same size. Thereby, we guarantee that there are pairs of tasks that are similar to each other. In Figure 3 (left), each color represents the partition of the original task and each ellipse represents the mean and four times the standard deviation of the latent task representations. One can see that the closest neighbour of each task is the other task that belongs to the same original task.

The second experiment targets multi-fidelity problems that arise when training a machine learning model on large datasets and approximate versions of the target objective are generated by considering subsamples of different sizes. We used the SVM surrogate for different dataset subsets from Klein et al. [25], which consists of a random forest trained on a grid of hyperparameter configurations of a SVM evaluated on different subsets of the training data. In particular, we defined the following subsets: $\{1/512, 1/256, 1/128, 1/64, 1/32, 1/16, 1/8, 1/4, 1/2, 1\}$ as tasks and sampled 100 configurations per task to train our meta-model. Note that we only provide the observed targets and not the subset size to our model. Figure 3 (right) shows the latent space of the trained meta-model: the latent representation of the model captures that similar data subsets are also close in the latent space. In particular, the first latent dimension $h_0$ coherently captures the sample size, which is learned using exclusively the correlation between the datasets and with no further information about their size.

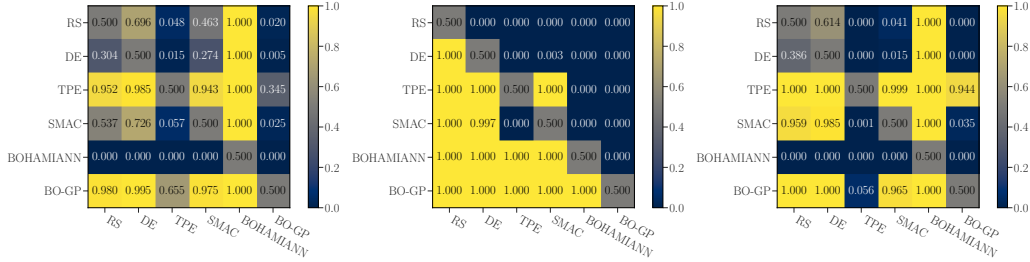

Figure 4: Heatmaps of the p-values of the pairwise Mann-Whitney U test on three scenarios. Small p-values should be interpreted as finding evidence that the method in the column outperforms the method in the row. Using tasks from our meta-model lead to results that are close to using the large set of original tasks. *Left*: results with 1000 real tasks. *Middle*: subset of only 9 reals tasks. *Right:* results with 1000 tasks generated from our meta-model.

## 5.2 Benchmarking with PROFET

Comparing HPO methods using a small number of instances affects our ability to properly perform statistical tests. To illustrate this we consider a distribution of tasks that are variations of the Forrester function $f(x) = ((\alpha x - 2)^2) \sin(\beta x - 4)$. We generated 1000 tasks by uniformly sampling random $\alpha$ and $\beta$ in $[0, 1]$ and compared six HPO methods: RS, DE, TPE, SMAC, BOHAMIANN and BO-GP (we left CMA-ES out because the python version does not support 1-dimensional optimization problems).

Figure 4 (left) shows the p-values of all pairwise comparisons with the null hypothesis 'Method$_{column}$ achieves a higher error after 50 function evaluations averaged over 20 runs than Method$_{row}$' for the Mann-Whitney U test. Squares in the figure with a p-value smaller than $0.05$ are comparisons in which with a 95% confidence we have evidence to show that the method in the column is better that the method in the row (we have evidence to reject the null hypothesis). To reproduce a realistic setting where one has access to only a small set of tasks, we picked 9 out of the 1000 tasks randomly. Now, in order to acquire a comparable number of samples to perform a statistical test, we performed 2220 runs of each method on every task, and then computed the average of groups of 20 runs, such that we obtained 999 samples per method to compute the statistical test. One can see in Figure 4 (middle), that although the results are statistically significant, they are misleading: for example, BOHAMIANN is dominating all other methods (except BO-GP), whereas it is significantly worse than all other methods if we consider all 1000 tasks.

To solve this issue and obtain more information from the same limited number of a subset of 9 tasks, we use PROFET. We first train the meta-model on the same 9 selected tasks and then use it to generate 1000 new surrogate tasks (see Appendix C for a visualization). Next, we use these tasks to run the comparison of the HPO methods. Results are shown in Figure 4 (right). The heatmap of statistical comparisons reaches very similar conclusions to those obtained with the original 1000 tasks, contrary to what happened when we did the comparisons with 9 tasks only (i. e. p-values are closer to the original ones). We conclude that our meta-model captures the variability across tasks such that using samples from it (generated based on a subset of tasks) allows us to draw conclusion that are more in line with experiments on the full dataset of tasks than running directly on the subset of tasks.

## 5.3 Comparing State-of-the-art HPO Methods

We conducted 20 independent runs for each method on every task of all three problem classes described in Section 4.1 with different random seeds. Each method had a budget of 200 function evaluations per task, except for BO-GP and BOHAMIANN, where, due to their computational overhead, we were only able to perform 100 function evaluations. Note that conducting this kind of comparison on the original benchmarks would have been prohibitively expensive. In Figure 5 we show the ECDF curves and the average ranking for the noiseless version of the SVM benchmark. The results for all other benchmarks are shown in Appendix E. We can make the following observations:

- Given enough budget, all methods are able to outperform RS. BO approaches can exploit their internal model such that they start to outperform RS earlier than evolutionary algorithms (DE, CMA-ES). Thereby, more sophisticated models, such as Gaussian processes or

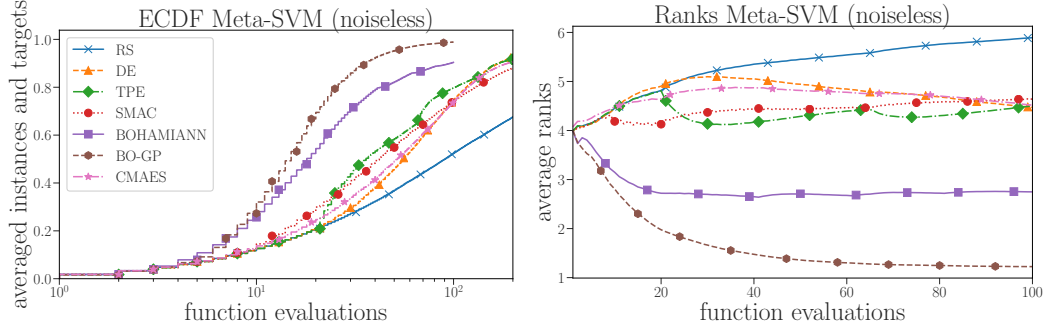

Figure 5: Comparison of various HPO methods on 1000 tasks of the noiseless SVM benchmark: *left:* ECDF for the runtime *right*: average ranks. See Appendix D for the results of all benchmarks.

> Bayesian neural networks are more *sample efficient* than somewhat simpler methods, e. g. random forests or kernel density estimators.

- The performance of BO methods that model the objective function (BO-GP, BOHAMIANN, SMAC) instead of just the distribution of the input space (TPE) decays if we evaluate the function through noise. Also evolutionary algorithms seem to struggle with noise.

- Standard BO (BO-GP) works superior on these benchmarks, but its performance decays rapidly with the number of dimensions.

- Runner-up is BOHAMIANN which works slightly worse than BO-GP but seems to suffer less under noisy function values. Note that this result can only be achieved by using PROFET as we could not have evaluated with and without noise on the original datasets.

- Given a sufficient budget, DE starts to outperform CMA-ES as well as BO with simpler (and cheaper) models of the objective function (SMAC, TPE), making it a competitive baseline particularly for higher dimensional benchmarks.

# 6   Discussion and future work

We presented PROFET, a new tool for benchmarking HPO algorithms. The key idea is to use a generative meta-model, trained on offline generated data, to produce new tasks, possibly perturbed by noise. The new tasks retain the properties of the original ones but can be evaluated inexpensively, which represents a major advance to speed up comparisons of HPO methods. In a battery of experiments we have illustrated the representation power of PROFET and its utility when comparing HPO methods in families of problems where only a few tasks are available.

Besides these strong benefits, there are certain drawbacks of our proposed method: First, since we encode new tasks based on a machine learning model, our approach is based on the assumptions that come with this model. Second, while we show in Section 5 empirical evidence that conclusions based on PROFET are virtually identical to the ones based on the original tasks, there are no theoretical guarantees that results translate one-to-one to the original benchmarks. Nevertheless, we believe that PROFET sets the ground for further research in this direction to provide more realistic use-cases than commonly used synthetic functions, e. g. Branin, such that future work on HPO can rapidly perform reliable experiments during development and only execute the final evaluation on expensive real benchmarks. Ultimately, we think this is an important step towards more reproducibility, which is paramount in such a empirical-driven field as AutoML.

A possible extension of PROFET would be to consider multi-fidelity benchmarks [25, 23, 27, 29] where cheap, but approximate fidelities of the objective function are available, e. g. learning curves or dataset subsets. Also, different types of observation noise, e.g non-stationary or heavy-tailed distributions as well as higher dimensional input spaces with discrete and continuous hyperparameters could be investigated. Furthermore, since PROFET also provides gradient information, it could serve as a training distribution for learning-to-learn approaches [6, 42].

## Acknowledgement

We want to thank Noor Awad for providing an implementation of differential evolution.

## Footnotes

[1]Note that we model an homoscedastic noise, because of that, $\hat{\sigma}_{\boldsymbol{\theta}_i}^2$ does not depend on the input

[2]We used the implementation from Chen and Guestrin [4]

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
