[Supplementary Material · appendix.pdf]

# Meta-Surrogate Benchmarking for Hyperparameter Optimization

**Aaron Klein**[1]  **Zhenwen Dai**[2]  **Frank Hutter**[1]  **Neil Lawrence**[3]  **Javier González**[2]
[1]University of Freiburg    [2]Amazon Cambridge    [3]University of Cambridge
{kleinaa,fh}@cs.uni-freiburg.de
{zhenwend, gojav}@amazon.com
ndl21@cam.ac.uk

## A    Hyperparameter Optimization Benchmarks

In Table 1 we list all OpenML datasets that we used to generate the Meta-SVM and Meta-FCNet benchmarks and in Table 2 the UCI datasets that we used for the Meta-XGBoost benchmark. The ranges of the hyperparameters for all benchmarks are given in Table 3. Figure 1 shows the empirical cumulative distribution over the observed target values based on the Sobol grid for all tasks.

| Name | OpenML Task ID | number of features | number of datapoints |
|---|---|---|---|
| kr-vs-kp | 3 | 37 | 3196 |
| covertype | 2118 | 55 | 110393 |
| letter | 236 | 17 | 20000 |
| higgs | 75101 | 29 | 98050 |
| optdigits | 258 | 65 | 5620 |
| electricity | 336 | 9 | 45312 |
| magic telescope | 75112 | 12 | 19020 |
| nomao | 146595 | 119 | 34465 |
| gas-drift | 146590 | 129 | 13910 |
| mfeat-pixel | 250 | 241 | 2000 |
| car | 251 | 7 | 1728 |
| churn | 167079 | 101 | 1212 |
| dna | 167202 | 181 | 3186 |
| vehicle small | 283 | 19 | 846 |
| vehicle | 75191 | 101 | 98528 |
| MNIST | 3573 | 785 | 50000 |

Table 1: OpenML dataset we used for the FC-Net and SVM classification benchmarks

Figure 1: The empirical cumulative distribution plots of all observed target values for all tasks.

| Name | number of features | number of datapoints |
|---|---|---|
| boston housing | 13 | 506 |
| concrete | 9 | 1030 |
| parkinsons telemonitoring | 26 | 5875 |
| combined cycle power plant | 4 | 9568 |
| energy | 8 | 768 |
| naval propulsion | 16 | 11934 |
| protein structure | 9 | 45730 |
| yacht-hydrodynamics | 7 | 308 |
| winequality-red | 12 | 4898 |
| slice localization | 386 | 53500 |

Table 2: UCI regression dataset we used for the XGBoost benchmark. All dataset can be found at https://archive.ics.uci.edu/ml/datasets.html

| | Name | Range | log scale |
|---|---|---|---|
| SVM | $C$ | $[e^{-10}, e^{10}]$ | ✓ |
| | $\gamma$ | $[e^{-10}, e^{10}]$ | ✓ |
| FC-Net | learning rate | $[10^{-6}, 10^{-1}]$ | ✓ |
| | batch size | $[8, 128]$ | ✓ |
| | units layer 1 | $[16, 512]$ | ✓ |
| | units layer 2 | $[16, 512]$ | ✓ |
| | drop. rate l1 | $[0.0, 0.99]$ | - |
| | drop. rate l2 | $[0.0, 0.99]$ | - |
| XGBoost | learning rate | $[10^{-6}, 10^{-1}]$ | ✓ |
| | gamma | $[0, 2]$ | - |
| | L1 regularization | $[10^{-5}, 10^{3}]$ | ✓ |
| | L2 regularization | $[10^{-5}, 10^{3}]$ | ✓ |
| | number of estimators | $[10, 500]$ | - |
| | subsampling | $[0.1, 1]$ | - |
| | max. depth | $[1, 15]$ | - |
| | min. child weight | $[0, 20]$ | - |

Table 3: Hyper-parameter configuration space of the support vector machine (SVM), fully connected neural network (FC-Net) and the gradient tree boosting (XGBoost) benchmark.

## B Comparison Random Search vs. Bayesian Optimization on XGBoost

For completeness we show in Figure 2 the comparison of random search (RS) and Bayesian optimization with Gaussian processes (BO-GP) on several UCI regression datasets. Out of the 10 datasets, GP-BO perform better than RS on 7, worse on one, and ties on 2 and hence performs overall better than RS which is inline with the results obtained from out meta-model. However, if we would look only on the first three datasets: Boston-Housing, PowerPlant and Concrete it would be much harder to draw strong conclusions.

## C Details about the Forrester benchmark

Figure 3 shows the original 9 tasks (left), their representation on the latent space of the model (middle) and an example of 10 new generated task (right), that resemble the original ones.

# D    Samples for the Meta-SVM benchmark

In Figure 4 and Figure 4 we show additional randomly sampled tasks with and without noise. One can see that, while the general characteristics of the original objective function, i.e. bowl shaped around the lower right corner, remains, the local structure changes across samples.

# E    Comparison of HPO Methods

We now described the specific details of each optimizer in turn.

**Random search** (RS) [1] We defined a uniform distribution over the input space and, in each iteration, randomly sampled a datapoint from this distribution.

**Differential Evolution** (DE) [9] maintains a population of data points and generates new candidate points by mutation random points from this population. We defined the probability for mutation and crossover to be 0.5. The population size was 10 and we sampled new candidate points based on the 'rand/1/bin' strategy.

**Tree Parzen Estimator** (TPE) [2] is a Bayesian optimization method that uses kernel density estimators (KDE) to model the probability of 'good' points in the input space, i e. that achieve a function value that is lower than a certain value and 'bad' points that achieve a function value smaller than a certain value. TPE selects candidates by maximizing the ration between the likelihood of the two KDEs which is equivalent to optimizing expected improvement. We used the default hyperparameters provided by the hyperopt (`https://github.com/hyperopt/hyperopt`) package.

**SMAC** [6] is a Bayesian optimization method that uses random forests to model the objective function and stochastic local search to optimize the acquisition function. We used the default hyperparameters of SMAC, and set the number of trees for the random forest to 10.

**CMA-ES** [5] is an evolutionary strategy that models a population of points as a multivariate normal distribution. We used the open source pycma package (`https://github.com/CMA-ES/pycma`). We set the initial standard deviation of the normal distribution to 0.6.

**Gaussian Process based Bayesian optimization** (BO-GP) as described by Snoek et al. [7]. We used expected improvement as acquisition function and an adapted random search strategy to optimize the acquisition function, which given a maximum number of allowed points $N = 500$ samples first 70% uniformly at random and the rest from a Gaussian with a fixed variance around the best observed point. While other methods, such as gradient ascent techniques or continuous global optimization methods could also be used, we found this to work faster and more robustly. We marginalized the acquisition function over the Gaussian process hyperparameters [7] and used the emcee package (`http://dfm.io/emcee/current/`) to sample hyperparameter configurations from the marginal log-likelihood. We used a Matern 52 kernel for the Gaussian process.

**BOHAMIANN** [8] uses a Bayesian neural network inside Bayesian optimization where the weights are sampled based on stochastic gradient Hamiltonian Monte-Carlo [3]. We use a step length of $10^{-2}$ for the MCMC sampler and increased the number of burnin step by a factor of 100 times the number of observed data points. In each iteration, we sampled 100 weight vectors over 10000 MCMC steps. We used the same random search method to optimize the acquisition function as for BO-GP.

All methods started from a uniformly sampled point and we estimated the incumbent after each function evaluation as the point with the lowest observed function value.

In Figure 6 and Table 4 we show the aggregated results based on the runtime and the ranking for all methods on all three benchmarks. We also show in Figure 6 the p-values of the Mann-Whitney U test between all methods. For a detailed analysis of the results see Section 5.3 in the main paper.

# F    Details of the Meta-Model

The neural network architecture for our meta-model consisted of 3 fully connected layers with 500 units each and tanh activation functions. The step length for the MCMC sampler was set to $10^{-2}$ and we used the first 50000 steps as burn-in. For the probabilistic encoder, we used Bayesian

| Benchmark | RS | DE | TPE | SMAC | BOHAMIANN | CMAES | BO-GP |
|---|---|---|---|---|---|---|---|
| Meta-SVM (noiseless) | 52.19 | 74.37 | 79.64 | 73.77 | 90.33 | 73.69 | **98.88** |
| Meta-SVM (noise) | 56.64 | 77.29 | 76.44 | 78.56 | **89.80** | 76.27 | 88.70 |
| Meta-FCNet (noiseless) | 45.71 | 77.99 | 78.73 | 72.71 | 82.50 | 56.31 | **84.71** |
| Meta-FCNet (noise) | 33.66 | 49.88 | 46.84 | 43.09 | **57.28** | 37.41 | 56.04 |
| Meta-XGBoost (noiseless) | 41.59 | 80.35 | 71.02 | 84.95 | 94.01 | 77.17 | **94.69** |
| Meta-XGBoost (noise) | 41.71 | 80.05 | 71.05 | 85.34 | 94.23 | 77.15 | **94.87** |
| Meta-SVM (noiseless) | 5.89 | 4.47 | 4.50 | 4.64 | 2.75 | 4.52 | **1.22** |
| Meta-SVM (noise) | 5.72 | 4.13 | 4.42 | 4.11 | **2.62** | 4.17 | 2.84 |
| Meta-FCNet (noiseless) | 5.67 | 3.70 | 3.72 | 4.09 | 2.90 | 5.14 | **2.79** |
| Meta-FCNet (noise) | 4.92 | 3.74 | 3.95 | 4.26 | **3.21** | 4.66 | 3.27 |
| Meta-XGBoost (noiseless) | 6.15 | 4.11 | 4.95 | 3.78 | 2.40 | 4.57 | **2.03** |
| Meta-XGBoost (noise) | 6.15 | 4.12 | 4.96 | 3.76 | 2.39 | 4.58 | **2.02** |

Table 4: *Top*: Each element of the table shows the averaged runtime after 100 function evaluations for each method-benchmark pair. *Bottom*: Same but for the ranking of the methods.

GP-LVM[1][10] with a Matern52 kernel to learn a $Q = 5$ dimensional latent space for the task description.

# G   Pseudo Code for Profet

Algorithm 1 shows pseudo code to evaluate an algorithm $\alpha$ with Profet by sampling new surrogate tasks (see Algorithm 2) sampled from our meta-model.

---

**Algorithm 1** Evaluating the performance of HPO methods.

*Inputs*: Datasets $\mathcal{D}_t = \{(\boldsymbol{x}_{ti}, y_{ti})\}_{i=1}^{N_t}$ for $t = 1, \ldots, T$ tasks. HPO method $\alpha$. Number of tasks $M$.

---

Train the probabilistic encoder $p(\mathbf{h}_t \mid \boldsymbol{y})$ and multi-task model $p(\theta|\mathcal{D})$ as described in Section 3.2 on dataset $\mathcal{D}$

Sample $M$ tasks using Algorithm 2

Solve $t_1, \ldots, t_M$ using $\alpha$ and compute $r(\alpha, t_m)$

Approximate $S_p(\alpha)$ in Equation 1 using $\frac{1}{M} \sum_{m=1}^{M} r(\alpha, t_m)$.

---

---

**Algorithm 2** Sampling new tasks.

*Inputs*: $noiseless \in \{true, false\}$, encoder $p(\mathbf{h}_t \mid \boldsymbol{y})$ and multi-task model $p(\theta|\mathcal{D})$ as described in Section 3.

---

Sample latent task vector $\mathbf{h}_{t_\star} \sim p(\mathbf{h}_t \mid \boldsymbol{y})$.

Sample a set of weights $\theta \sim p(\theta|\mathcal{D})$ from the posterior of the BNN.

**if** $noiseless == true$ **then**

$f_{t_\star}(\boldsymbol{x}) = \hat{\mu}(\boldsymbol{x}, \mathbf{h}_{t_\star}|\theta)$

**else**

$f_{t_\star}(\boldsymbol{x}) = \hat{\mu}(\boldsymbol{x}, \mathbf{h}_{t_\star}|\theta) + \epsilon \cdot \sigma(\boldsymbol{x}, \mathbf{h}_{t_\star}|\theta)$ where $\epsilon \sim \mathcal{N}(0, 1)$

**end if**

**return** $f_{t_\star}(\boldsymbol{x})$

---

## Footnotes

[1]We used the implementation from GPy [4]

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

Figure 2: Comparison of Bayesian optimization with Gaussian processes (GP-BO) and random search (RS) for optimizing the hyperparameters of XGBoost on 10 UCI regression datasets.

Figure 3: Visualizing the concept of our meta-model on the one-dimensional Forrester function. *Left:* 9 different tasks (solid lines) coming from the same distribution. *Middle:* We use a probabilistic encoder to learn a two-dimensional latent space for the task embedding. *Right:* Given our encoder and the multi-task model we can generate new tasks (dashed lines) that, based on the collected data, resemble the original tasks.

Figure 4: Noisy samples from our meta-model for the SVM benchmark

Figure 5: Noiseless samples from our meta-model for the SVM benchmark

Figure 6: Comparison of various different methods on all three HPO problems. From above to below 2-dimensional support vector machine, 6-dimensional feed-forward neural network and 8-dimensional XGBoost. The first column shows the ECDF, the second column the ranking and last column the p-values of the Mann-Whitney U test for the noisy and noiseless version of each HPO problem.