[Reviews · NeurIPS 2019]

Reviewer 1



The motivation of this paper is clear. That is reducing the evaluation cost on the hyper-parameter tuning tasks. Some previous works have been proposed to tackle this issue, such as warm-starting optimization processes, multi-fidelity optimization, etc. This paper proposed to build a meta-model among problems that are from a problem distribution. By applying the Gaussian processing to capture the relations among problems (it likes the surrogate function in Bayesian optimization), the evaluations in the new problems are cheaper. However, this paper is poorly-written and poorly-organized. There are full of typos and grammatical issues in this paper. Some detail comments are listed below: 1. In Figure 1, how much hyper-parameters are you selected in the experiments? Random search is hard to beat GP-BO when hyper-parameters are more than 2 in common sense. 2. What is your algorithm framework of the proposed method? Can you show the pseudo-code of the proposed method? 3. What is the conclusion we can get from the Figure 3? Can you explain it?

Reviewer 2



The idea of generating surrogate tasks or datasets for evaluating HPO methods is quite interesting, and is especially useful if the target task or model is very computationally expensive to evaluate a hyperparameter configuration, e.g., ResNet. Unfortunately, the empirical evaluations fail to effectively support the superiority and feasibility of the proposed benchmarking algorithm in practical applications. --------------------------------------------------------------------------------------------------------- I have read the authors' response, and tend to maintain my score. The authors have addressed some of my concerns, while I am even more worried about the feasibility and practicability of the proposed benchmarking algorithm (cf. Line 30-32).

Reviewer 3



Originality: Conditioning on data for existing tasks to generate an unbounded number of tasks appears to be a novel idea. Quality: The paper is very well written. The idea is clear and well motivate. The idea could also be very useful to advance reproducibility in black-box optimization. The experiments are thorough and well executed. Clarity: As mentioned already, the paper is very well written. Significance: This work and in particular the tasks if released are bound to become useful benchmarks for folks working on black-box optimization.

[Author Response · NeurIPS 2019]

First of all, we thank all reviewers for their valuable time and feedback. Reflected in this years reviewing process, reproducibility is of central importance to the whole NeurIPS community and was also unanimously identified during the panel discussion at the AutoML workshop @ ICML19 as one of the major challenges the AutoML community has to face. Unfortunately, hyperparameter optimization (HPO) often requires tremendous computational resources which renders reproducibility hard in practice, since one can only afford a few function evaluations. As an important step to enable better reproducibility we provide a principled way to generate cheap-to-evaluate benchmarks which contain the typical characteristics of real HPO problems.

We thank the reviewers for pointing out typos and grammatical errors, which we of course have fixed now. We will not address these any further here and proceed by addressing the reviewers' comments in turn.

**R1:** We are afraid that the reviewer might have misunderstood some parts of the paper. The goal is **not** to speed up Bayesian optimization, such as warm-starting or multi-fidelity optimization, but instead to provide a cheap-to-evaluate and realistic **benchmark suite** for hyperparameter optimization methods. This allows the community to execute exhaustive experiments with a low computational budget and to easily compare to existing methods, which is necessary to make HPO more reproducible. We have made this claim clearer in a revised version of the manuscript to avoid confusion. We strongly believe that methods able to tackle the reproducibility problem are essential in modern machine learning, and we hope our clarifications will help the reviewer to support our contribution in this direction.

1. Figure 1 shows the XGBoost benchmark with 8 hyperparameters described in Section 4.1. Due to space constraints further details can be found in Appendix A (as referenced in the main paper).

2. In a nutshell, we first learn a latent embedding across optimization tasks together with a generative multi-task model that allows us to sample an infinite amount of new optimization tasks which resemble the original ones. We have added pseudocode to make the proposed method more tangible.

3. Figure 3 visualizes the learned latent space and shows that our embedding indeed captures similarities across tasks (see also Section 5.1 for further details).

**R2:** We thank reviewer 2 for the constructive feedback:

About the methodology: The way we learn the latent embedding is straightforward and follows the general GPLVM framework, which, given a matrix with all observed target values across all tasks, learns a latent Gaussian distribution for each task. We refer to the original paper for further details about the approximation of the variational posterior. The subscript n indicates the datapoint (where N is the total number of datapoint) and h indicates the sample drawn from the latent distribution over tasks provided by our embedding. We have made this more clear in the main paper now.

About the experiments: We would love to conduct the same analysis that we did for the Forrester function in Section 5.2 also for real HPO problems. However, this is (i) computationally impossible (and can only be conducted using Profet) and (ii) we do not have access to any HPO problem where 1000 real tasks (or datasets) are available.

The hyperparameters for BOHAMIANN (together with the hyperparameters for all other methods) are, due to space constraints, described in Appendix E and follow the default parameters proposed by Springenberg et al. Note that, consistent with our results, also in the original paper by Springenberg et al. BOHAMIANN was outperformed by BO-GP in low dimensional continuous problems (for example see Figure 1 in Springenberg et al.) and seems to improve upon BO-GP if the dimensionality increases.

About Section 5.2: the reason why results with 1000 generative tasks stick more to the result to 1000 original tasks than the subset of 9 tasks is because our generative model captures the variability of tasks. We added further details.

**R3:** We thank reviewer 3 for the helpful feedback and agree that the benchmarks will be key for researchers working in black-box or hyperparameter optimization. Indeed, it is surprising that the community hasn't yet produced a lot of research in this direction, ML being a discipline that is being applied in such a long list of real applications. Many thanks also to the proposed improvements, which we found very helpful. While we think they are out-of-scope for this paper, we actually plan to include them into future work.

1. Indeed, we also think complex search space are interesting and having a benchmark suite would enable future work to tackle these spaces.

2. We have added a discussion about PBT and Hyperband in the related work section. Note that, we are planning to extend our benchmarks to also model fidelities of the objective function in order to apply multi-fidelity methods, such as Hyperband or BOHB.

3. Having different kinds of noise is indeed a good idea. Since our multi-task model is a Bayesian neural network, it would be possible to adapt the likelihood and the predictive distribution to allow for other noise models, such as Student'T distributions.

[Meta-Review · NeurIPS 2019]

Reviewers are not entirely satisfied with your response, however, they are leaning to a positive overall opinion (except R1, but I tend to disagree a bit in her/his evaluation). Hence I think your paper can be accepted provided (and I am really trusting on you, as there is no way to obligue you) you commit to address in as much as possible the open issues raised by reviewers. I partially agree with the appreciation from R1 and that is why I am relying on the opinion from the other reviewers. Furthermore, I read your whole document and I do think it delivers a contribution that can be of interest to the community.